# Iridescent structural coloration in a crested Cretaceous enantiornithine bird from the Jehol Biota

Zhiheng Li[1]*[†], Jinsheng Hu[2][†], Thomas A Stidham[1,3], Mao Ye[2], Min Wang[1], Yanhong Pan[4], Tao Zhao[5], Jingshu Li[1], Zhonghe Zhou[1,6], Julia A Clarke[7]

[1]Key Laboratory of Vertebrate Evolution and Human Origins, Institute of Vertebrate Paleontology and Paleoanthropology, Chinese Academy of Sciences, Beijing, China; [2]School of Instrumentation and Optoelectronic Engineering, Beihang University, Beijing, China; [3]Department of Biology, Austin College, Sherman, Texas, United States; [4]State Key Laboratory of Critical Earth Material Cycling and Mineral Deposits, School of Earth Sciences and Engineering, Centre for Research and Education on Biological Evolution and Environment and Frontiers Science Center for Critical Earth Material Cycling, Nanjing University, Nanjing, China; [5]Institute of Palaeontology, Yunnan Key Laboratory of Earth System Science, Yunnan University, Kunming, China; [6]College of Earth and Planetary Sciences, University of Chinese Academy of Sciences, Beijing, China; [7]Department of Earth and Planetary Sciences, Jackson School of Geosciences, The University of Texas at Austin, Austin, Texas, United States

*For correspondence:
lizhiheng@ivpp.ac.cn

[†]These authors contributed equally to this work

Competing interest: The authors declare that no competing interests exist.

## eLife Assessment

This study presents a potentially **fundamental** analysis of a fossil feather from a 120-million-year-old enantiornithine bird. Using sophisticated 3D microscopic and numerical methods, the authors conclude that the feather was iridescent and brightly colored, possibly indicating that this was a male bird that used its crest in sexual displays. At present, the strength of evidence supporting the conclusions is considered **incomplete** based on methodological shortcomings and questions about taphonomy.

**Abstract** A combination of sectioning and microscopy techniques, along with the application of finite-difference-time-domain modeling on a fossil feather, results in the novel estimation of the range of iridescent colors from the fossilized melanosome type and organization preserved in the elongate head crest feathers of a new Cretaceous enantiornithine bird. The densely packed rod-like melanosomes are estimated to have yielded from red to deep blue iridescent coloration of the head feathers. The shape and density of these melanosomes also may have further increased the feather's structural strength. This occurrence on a likely male individual is highly suggestive of both a signaling function of the iridescent crest and a potential behavioral role in adjusting the angle of light incidence to control the display of this iridescent structural coloration.

## Introduction

The recognition and study of melanosomes in fossil feathers of non-avialan dinosaurs and birds has added greater complexity to the pattern of their diversity and macroevolution, demonstrating the intertwined evolution of coloration with feather functions across display, thermoregulation, and

other uses (*Li et al., 2012*; *Li et al., 2014*; *Terrill and Shultz, 2023*). This linkage among colors, structures, and functions early in feather evolution extends even to convergence in iridescent feathers among non-avialan theropods, and it is highly suggestive of commonalities shared among stem and crown birds in terms of their use of colored pigments for camouflage, structural strength, and display (*Hu et al., 2018*; *Foth and Rauhut, 2020*). In a comparative setting with the melanosome feather coloration of living birds, workers have endeavored to reconstruct the colors of fossil feathers using quantitative means focused predominantly on melanosome geometry and density, and the results point to the fossilized melanosomes having produced patterns of black, brown, rufous, gray, and other colors (*Vinther et al., 2008*; *Li et al., 2010*; *Zhang et al., 2010*; *Li et al., 2012*; *Peteya et al., 2017*; *Benton et al., 2019*). Furthermore, the specific shape and arrangement of some fossilized feather melanosomes derived from both non-avialan theropod dinosaurs and birds are similar to those required for producing structurally based iridescent feather coloration in living birds, a case of independent evolution (*Stoddard and Prum, 2011*; *Hu et al., 2018*; *Pan et al., 2022*).

Feathers serve multiple functions in birds from comprising an insulatory epidermal layer to forming aerodynamic flight surfaces, and as structures for the containment and display of various pigments. Living birds exhibit diverse feather morphologies and colors, and the significant diversity of known fossil feathers, along with their preserved pigments, points to their past use for camouflage, as well as display and signaling (*Prum, 2006*; *Foth and Rauhut, 2020*; *Terrill and Shultz, 2023*). While the display of most colored feathers is passive and available to any viewer, birds can behaviorally control their signaling through: the erection of feathered ornaments such as the head crest in hoopoes (*Upupa*) or peacock (*Pavo*) tails; the exposure of normally hidden brightly colored crown feathers as in kinglets (*Regulus*); and even the refined control of the black and iridescent coloration in birds such as hummingbirds (Trochilidae) and birds of paradise (Paradisaeidae; *Zi et al., 2003*; *Wilts et al., 2014*; *Stoddard et al., 2020*). This great variation in structural or iridescent colors results from differences in pigments, their organization, and other structural modifications. For example, the variety of mixed structural colors present in a peacock feather is achieved through changes in the spacing in the keratinous matrix and melanosome organization (*Zi et al., 2003*). In addition to the normal range of spectral coloration, hummingbirds can see non-spectral colors as well, and their utility is supported by vision perception experiments (*Stoddard et al., 2020*). The interplay of bird of paradise feather coloration and their vision similarly indicates coordination of their mating displays with spectral properties through the avian visual system (*Wilts et al., 2014*). Individual birds actively control the signaling functions of their feathers with feathers or ornaments hidden/displayed or even the color presented (as dark/black vs. vibrant/iridescent) being consciously restricted or directed to particular viewers such as potential mates or rivals (*Wilts et al., 2014*) .

Fossilized feathers typically have been evaluated within two-dimensional surface views, limiting the understanding of their overall ultrastructure and perhaps their true coloration. As a step forward in our reconstruction of the structure and function (color display) of early bird feathers, we examined the unique head-crest feathers of a well-preserved specimen (Institute of Vertebrate Paleontology and Paleoanthropology, IVPP V26899) of the Early Cretaceous (~120 Ma) enantiornithine *Shangyang* sp. from the Jiufotang Formation of northeastern China (*Figure 1*). This fossil preserves a diverse feather assemblage across the body with dense contour feathers, flight remiges, and two rachis-dominated tail feathers (*Figure 1—figure supplement 1*). Among its diverse feather morphologies, this specimen also displays a strikingly enlarged feathered crest on the head that uniquely extends from the caudal end of the skull to a position rostral to the nasal bone (*Figure 1—figure supplement 1*). For our study, we sampled a long isolated mature feather that is displaced from the caudal part of the crest, as well as three feather fragments extracted from the in-situ position of the head crest.

In this study, we utilized serial ultrathin histological sectioning, scanning/transmission electron microscopy (S/TEM) imaging, and finite-difference time-domain (FDTD) modeling to investigate the potential iridescent coloration mechanisms in the elongated crest feathers from this newly discovered enantiornithine specimen of the Jehol Biota. Differing from previous statistical reconstructions of fossil color that measured melanosome aspect ratios, our new approach takes the three-dimensional shape and packing patterns of melanosomes, as well as the keratinous matrix, into consideration. With these novel modeling approaches, a more accurate coloration estimation was produced for its crest feather.

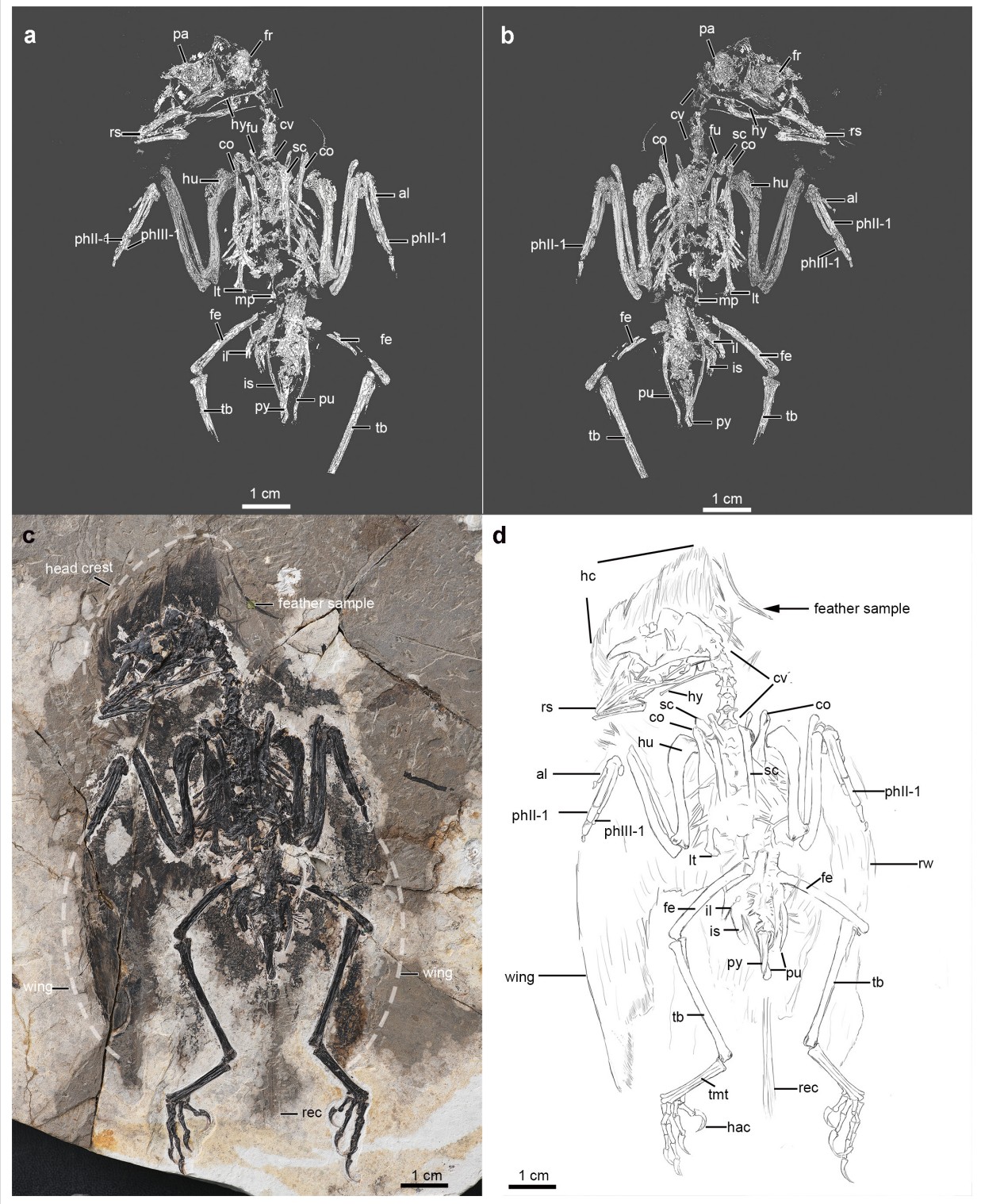

**Figure 1.** Digital rendering, photography, and line drawing of the enantiornithine fossil specimen (*Shanyang* sp., IVPP V26899) (**a**), (**c**), (**d**) - dorsal; (**b**) - ventral view showing the major skeletal elements. The feather extracted from the crown is labeled as a feather sample (**c**). Abbreviations: al, alular metacarpal; co, coracoid; cv, cervical vertebra; fe, femur; fi, fibula; fr frontal; fu, furcula; hac, hallucal ungual; hc, head crest; hu, humerus; hy, hyoid; il, ilium; is, ischium; lt, lateral trabecula; mcf, molted crest feather; mc II, major metacarpal; mp, midline process; ph II-1, first digit of phalanx II; phIII-1, first digit of phalanx III; pa, parietal; pu, pubis; py, pygostyle; r, radius; rec, rectrices; rem, remiges; rs, rostrum; rw, right wing; sc, scapula; sk, skull; st, sternum; tb, tibiotarsus; and tmt, tarsometatarsus.

*Figure 1 continued on next page*

*Figure 1 continued*

The online version of this article includes the following figure supplement(s) for figure 1:

**Figure supplement 1.** Photograph showing details of skeletal and integumentary anatomy of the new enantiornithine specimen (IVPP V26899): (**a**)-skull, (**b**)-pelvis and hindlimb, and (**c**)-pectoral region.

**Figure supplement 2.** Microscopic ground section images to show the osteo-histological features of the femur sampled (**a**)-overall view; (**b**) enlarged view labeled in black box.

## Results

In the new fossil, head crest feathers were selected from the caudal side of the head and sectioned using ultrathin serial sectioning. The overall shape of the sampled crest feather is well-preserved and comprised of elongate and relatively wide barbs. The dominant barb length is proportionally longer than the feather barb examined from the neck and head region of *Confuciusornis* (*Foth, 2012*). The feather is different from the bilaterally symmetric plumulaceous feathers of living birds, and we define it as a barb-dominant feather, since most of the exposed feather tissue is composed of long barbs with a very short and thick calamus. It is estimated to correlate with evolutionary stage II and IIIa of *Prum and Brush, 2002*, similar to the intermediate feather type proposed as found in an amber fossil (*Perrichot et al., 2008*). The new feather described here is distinct from the amber feather in showing an asymmetric shape, radically diverging toward one side (caudal side), and is much larger in size

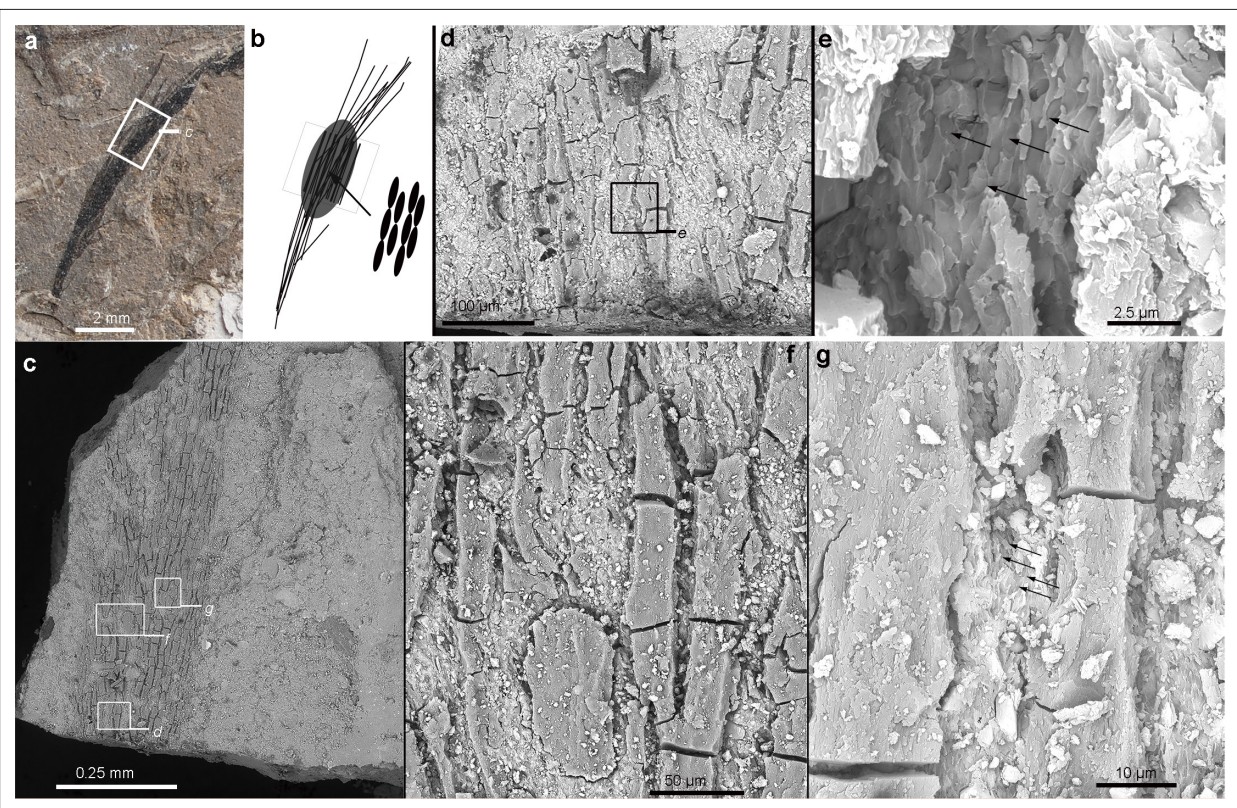

**Figure 2.** Photograph (**a**) and simplified line diagram (**b**) of the sampled isolated long feather with a focused area from SEM imaging (**c–g**). As shown in the SEM images, the long elliptical or oval-shaped melanosomes are tightly aligned with their long axis parallel to the elongated barbs (**e, g**). The position of enlarged images (**d**, **e**, **f**, and **g**) is indicated as square in (**c, d**); black arrows in (**e**) and (**g**) indicate the melanosome clusters.

The online version of this article includes the following figure supplement(s) for figure 2:

**Figure supplement 1.** SEM of the serial zoomed view (a to c) to show the lower portion of the sampled crest feather below the sectioned fragment.

**Figure supplement 2.** STEM images (**b–g**) of cross-sections taken from three different positions (indicated by white dashed lines in a) demonstrate similar melanosome packing styles.

**Figure supplement 3.** STEM images showing melanosome structure from three fragments of the feather crest (indicated by dashed lines and a white box in a) reveal the hooked linkages between melanosomes and their surrounding melanosome structures in (**b**), (**c**), and (**d**).

(*Perrichot et al., 2008*). There are dense clusters inside the barbs as seen from images of both SEM and TEM.

Ultrathin histological sectioning and application of SEM and S/TEM imaging of the fossil feather samples (see Appendix 1) resulted in the identification of partially interlinked melanosomes with a derived asymmetric packing pattern (*Figures 2 and 3*). The long axial crack in *Figure 2* may represent the surface of the barbs, resembling a crack formed in maturation experiments (*Zhao et al., 2020*). Those melanosome clusters are present and concentrated mostly within the barbs rather than in the barbules of extant bird feathers based on similar size (*Figure 2*). The long axis of the densely packed melanosomes is aligned overall parallel with the barb's long axis (*Figure 2*), which has been confirmed by detailed histological data as well. The melanosomes are arranged in an asymmetric hexagonal conformation, clearly visible in the different sections (see *Figure 3*).

SEM imaging of the cross section and surface view of the sampled feather indicates the presence of a few major barbs without barbules (*Figure 2—figure supplements 1–3*). The thickness of the complete feather barb with filled melanosomes (preserved) is around 10 μm, which is much greater than the typical diameter of known barbules, but similar to barbs in extant birds (*Freyer et al., 2021*). The three-dimensionally packed melanosomes were studied in both cross and nearly longitudinal sections within the dark-colored barbs (one in the middle of the barbs and another in the lateral side of the feather block, *Figure 3a and b*).

Hooks are commonly observed on the oval-shaped melanosomes in cross-sectional views, with two dominant types identified on the dorsal and ventral sides (*Figure 3c–d*, red arrows). These hooks are deflected in opposing directions, linking melanosomes from different arrays (dorsal-ventral) together. The major axis (y) of the oval-shaped melanosomes (mean = 283 nm) is oriented toward the left side in cross-section, while the shorter axis (x) measures approximately 186 nm (*Appendix 1—table 1*). In oblique or near-longitudinal sections (*Figure 3e–f*), the hooked structures' connections to the distal and proximal sides of neighboring melanosomes are clearly visible (blue arrows, *Figure 3f*). The mean long axis (z) of the melanosomes is approximately 1774 nm (*Appendix 1—table 1*). A similar pattern occurs in two additional regions of interest within the same feather (*Figure 2—figure supplement 2*). Although the smaller proximal hooks in these sections are less distinct, this may reflect developmental variation during melanosome formation along the feather barb. Significantly smaller hooks were also observed in cross-sections of in-situ feather barbs from the anterior side of the feather crest sampled (*Figure 2—figure supplement 3*). Based on these observations, we propose that the hooked structures—particularly those on the dorsal, ventral, proximal, and distal sides of the melanosomes—enhance the structural integrity of the barb (*Figure 3—figure supplements 1–2*). However, these features may be teratological and unique to this individual, as no similar structures have been reported in other species. These hooks may stabilize the stacked melanosome rods and contribute to increased barb dimensions, such as diameter and length. The sections exhibit modified (or asymmetric) hexagonally packed melanosomes with the presence of extra hooked linkages (*Figure 3c–d and e–f*). The long rod-like melanosomes are different from all other known feather melanosomes from both extant and extinct taxa in having some extra hooks and an oblique ellipse shape in cross and longitudinal sections of individual melanosomes (*Durrer, 1986*; *Zhang et al., 2010*). The asymmetric packing of the melanosomes (the major axis leans leftward in cross section) played a major role in the reduction of fossilized keratinous matrix within the barbs, which may correspond to a novel structural coloration in this extinct bird. The close-packed hexagonal melanosome pattern found in extant avian feathers yields rounded melanosome outlines in contrast to the oval-shaped melanosomes (see *Figure 4—figure supplement 1*, x<y) in the perpendicular section here. The asymmetric compact hexagonal packing (ACHP) of the melanosomes is different from the known pattern of melanosomes formed in the structure of barbules among extant birds (*Eliason and Shawkey, 2012*), which has been seen as a regular hexagonal organization. The packing of the melanosomes in an asymmetric pattern, on the microscopic level, might be related to the asymmetrical path of the barb extension direction observed at the macroscopic level (*Figure 2—figure supplements 1–3*).

In the computational (FDTD) simulation, when light is incident at a small angle (≤30°, both s- and p-polarized) in the X-Y plane, no evident reflection peak was found in the visible spectrum, as shown in *Figure 4c* (e.g. when a viewer observes at this angle, the light is almost completely absorbed, and the crest feather would have a darkened appearance). As the incident angle varies from 40° to 70°, the wavelength corresponding to the reflection peak shifts from 687 nm to 475 nm, and the color of

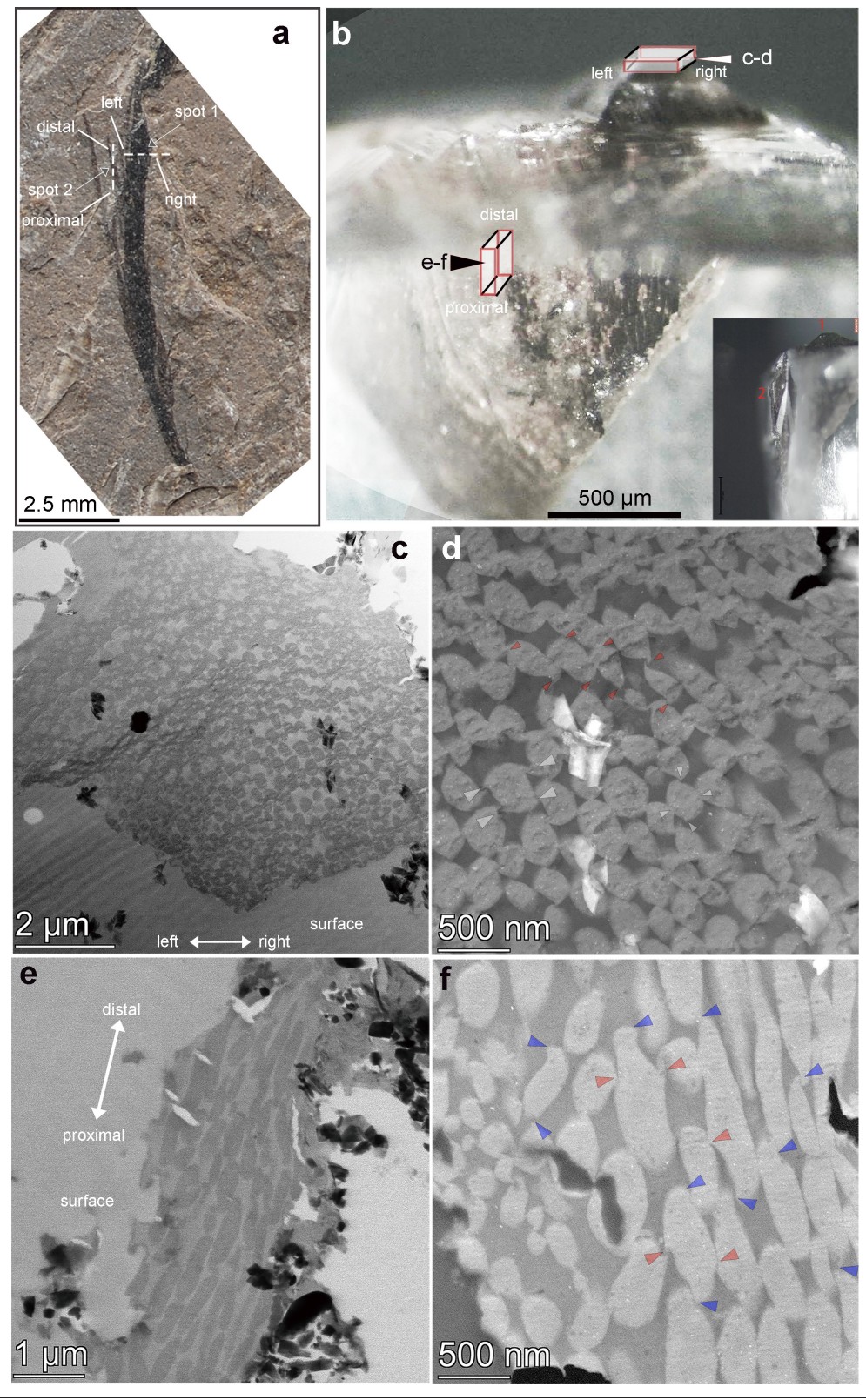

**Figure 3.** Photograph of the crest feather (**a**) and prepared blocks of the feather barb (**b**) for thin sectioning. The sampled spots for the sliced feather are indicated as long red-colored cubes with their respective positions in (**b**). (**c**) to (**f**) are (STEM) images obtained from individual slices as indicated in (**b**). The shape and arrangement of the melanosome packing (ACHP, asymmetric compact hexagonal packing) is clearly revealed in (**d**); (**c**) and (**d**) are

*Figure 3 continued on next page*

*Figure 3 continued*

derived from the cross-section; (**e**) and (**f**) are derived from the nearly longitudinal section. As indicated by the red arrows in (**d**) and (**f**), the dorsal and ventral hooks are dominant in the cross-sectional images. On the other hand, hooks connecting the proximal and distal arrays of melanosomes are better visualized in the longitudinal section, as labeled with blue arrows in (**f**). The orientation of the slices was labeled (left, right, proximal, distal) with their respective positioning in the fossil before and during sectioning.

The online version of this article includes the following figure supplement(s) for figure 3:

**Figure supplement 1.** Barb surface and cross- section morphology of the sampled fragment of the crest feather.

**Figure supplement 2.** Targeted feather barb block prepared in FIB-SEM, with volume rendering reconstruction based on the acquired sequential cross-sectional images; the volume reconstruction is visualized in the x-y plane (c-cross section view) and in x-z plane (d-sagittal section view).

the reflection peak gradually changes from red to deep blue (inset in *Figure 4c* clearly shows how the resulting color varies with the angle of incidence). From the perspective of optical modeling, the generation of iridescent color (reflection peaks in the visible spectrum) is caused by the constructive interference between scattered light produced by these uniquely shaped melanosomes that are compactly arranged in the barbs (*Vinther et al., 2010*). Between blue and red, reflection peaks of green and yellow also are present, corresponding to the wavelengths of 512 nm and 578 nm respectively. It is worth noting that in the case of both p- and s-polarized incident light, the positions of the reflection peaks are nearly identical, with only a slight difference in peak reflectance. Therefore, the three-dimensional multilayer nanostructure model is somehow equivalent to one-dimensional multilayer biphotonic crystals (*Saranathan et al., 2012*). In summary, the crest feather of the Cretaceous enantiornithine shows a strikingly angle-dependent and iridescent coloration pattern. When we consider the shrinkage effect on melanosomes demonstrated experimentally (*McNamara et al., 2013*) and various distances between melanosomes resulting from compression effects (*Pan et al., 2019*; *Zhao et al., 2020*), our additional FDTD modeling shows a similar reflection spectrum with a narrower bandwidth of the color peaks (*Figure 4—figure supplements 1–3*).

## Discussion

This study of a fossilized feather with a combination of microscopic imaging, ultrathin sectioning, and three-dimensional modeling has provided a more accurate examination of ancient structural coloration and uncovered unusual melanosome structures. Using previous conventional approaches alone, we would have classified this feather crest simply as black or close to iridescent based on its melanosome aspect ratio (*Nordén et al., 2021*). The recognition of overall iridescence links to a commonly found extant bird color range only known previously from the platelet-shaped dinosaurian melanosomes (*Hu et al., 2018*) and a few fossil feathers (*Pan et al., 2022*; *Nordén et al., 2019*; *Nordén et al., 2021*). The independent evolution of overall iridescence, similar to that found in living birds, was reconstructed here based on a novel array of melanosomes in a feather barb in an extinct non-crown group bird (*Nordén et al., 2019*; *Yoshioka and Akiyama, 2021*). The occurrence of iridescent structural coloration in an enlarged head crest on this ancient bird is similar to structures present across many living birds that employ iridescence for intimidation, signaling, and particularly for mate attraction or intraspecific territorial defense (*Terrill and Shultz, 2023*). Given that this individual bird likely was a male based on the occurrence of paired rachis dominant tail feathers (*Figure 1—figure supplement 1*), the vibrant coloration on a prominent feathered crest suggests that sexual selection may have long played an important role in avialan (feather) macroevolution (*Foth, 2020*; *Wang et al., 2021*). The vibrant color and its presence on an enlarged feather crest likely are interlinked and probably indicate an evolutionary and selective relationship among color, feather position, and feather morphology/size.

Our discovery of new melanosome shapes and networks suggests that birds evolved previously unrecognized patterns of color production outside of the limits present among crown group birds. Additionally, these pigments and their dense organization may have made the foundation of the feather darker and allowed for a more intense or vibrant iridescent coloration, a potential relationship that is understudied among living birds. While the interrelationship between melanin content in feathers and increases in structural strength have long been known (*Burtt, 1981*; *Burtt and Ichida,*

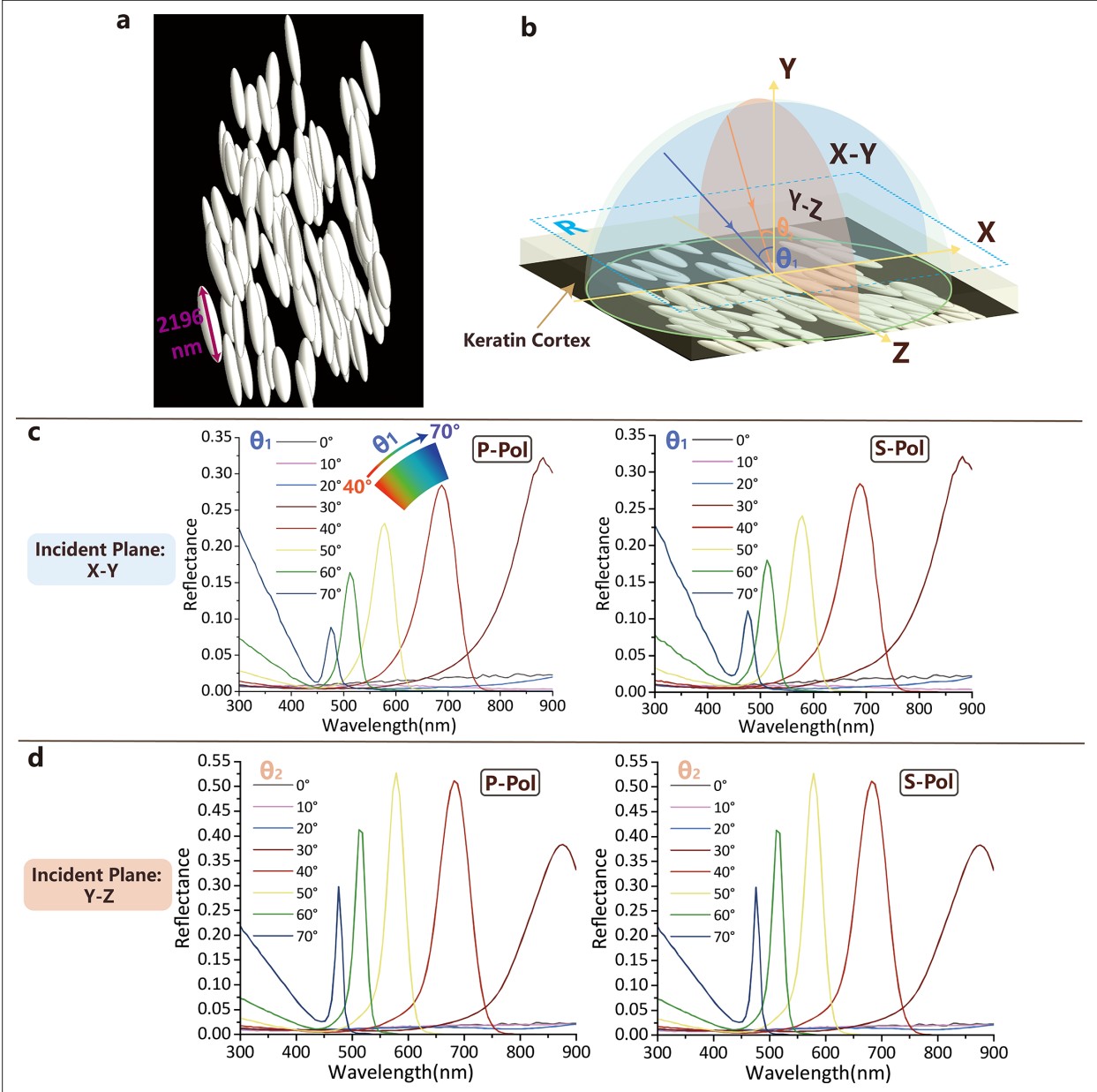

**Figure 4.** FDTD modeling results. (**a**) Representative multilayer melanosome model; (**b**) Schematic drawing of the reflectance calculation setup of the feather barb. X-Y incident plane (blue) is perpendicular, and the Y-Z incident plane (orange) is parallel to the special melanosome longitudinal axis. $\theta_1$ and $\theta_2$ are incident angles in X-Y and Y-Z incident planes, respectively. The blue dashed box (**R**) indicates the area where the reflection spectra are simulated and calculated. (**c**) Angle-dependent reflectance spectra calculated for p- and s-polarized light by FDTD modeling with X-Y incident plane. (**d**) Angle-dependent reflectance spectra calculated for p- and s-polarized light by FDTD modeling with Y-Z incident plane. The keratin cortex layer was considered in the modeling.

The online version of this article includes the following figure supplement(s) for figure 4:

**Figure supplement 1.** STEM images obtained from feather barb slices with the reconstructed 3D model melanosome sets.

**Figure supplement 2.** Refractive index dispersion.

**Figure supplement 3.** Comparison of angle-dependent reflectance spectra in the Y-Z incident plane for the normal and enlarged cases with consideration of melanosome shrinkage in taphonomy.

*2004*), perhaps the increased density of melanosomes in this fossil feather barbs added even more structural strength to the feathered crest of this bird in compromise for its lack of a dominant rachis. Additionally, the interlocking hooks among the melanosomes might have aided individual feather barb in resisting shear forces. The modified closely packed melanosome arrays reconstructed here also are potentially advantageous in increasing barb structural integrity and stability in the long barb ramus and likely affected the visualized spectrum during the display of iridescent coloration. We cannot completely rule out the increased melanosome density in the fossil feather as the result of tapho-nomy. Size reduction in melanosomes was found previously in a simulation experiment (*McNamara et al., 2013*), but such a reduction is very unlikely to result in the highly organized and consistent melanosome packing present here (*McNamara et al., 2013*; *Zhao et al., 2020*). To test the potential impact of melanosome shrinkage and potential taphonomic changes to their spacing, we conducted an additional analysis (See Appendix 1), and the results yield similar iridescent spectral peaks as the original setting.

Since the iridescent coloration of living birds is produced by the accumulation of melanosomes mostly positioned in barbules, the identified fossil barbs filled with melanosomes represent a unique phenotype related to an early feather evolutionary and developmental stage. Given the lack of barbules in most early primitive feather forms (or filamentary structures) in non-avialan dinosaurs and pterosaurs, the positioning of melanosomes in barbs might be a plesiomorphic feature for ornitho-dirans broadly (*Xu, 2019*; *Cincotta et al., 2022*). With different destinations of melanosome transport during feather formation, the pattern recovered here suggests fundamental differences in feather pigmentation among Cretaceous stem taxa as compared to crown birds. The melanosome-filled barbs evolved earlier in feather evolution before the later appearance of innovations such as the presence of a stiff rachis and colorful barbules observed across living feathers (*Yu et al., 2002*).

A focus on the two-dimensional examination of fossil feathers has uncovered a large number of black or dark melanosomes across many early birds and feathered theropods (*Li et al., 2014*; *Hu et al., 2018*). However, our three-dimensional study of a fossilized Jehol feather specimen demon-strates that ultra-histological detail and spatial relationships can be preserved at the microscopic level. Importantly, our discovery suggests that reexamination of previously published 'black' pigmented fossil feathers might uncover a wider occurrence of 'surprising' iridescence and structural coloration among these early birds and non-avialan taxa. Such discoveries will aid in documenting both inde-pendent and convergent evolutionary pathways in the history of birds. This research further demon-strates that Jehol fossil feather tissue can be treated as (microscopically) three-dimensional specimens allowing for the collection of valuable comparative data and application of comparative analytical techniques that facilitate a better understanding of early feather evolution and function.

Here, we recovered iridescent coloration in fossil bird feathers that was produced by three-dimensional melanosome packing patterns. This discovery dramatically expands the range of fossil bird coloration space, and this diversity in feather coloration in an early bird fits well with its forested environment, as proposed by the coeval fossil plant community (*Wu et al., 2023*). Tropical forest envi-ronments are accompanied by colorful flowering plants and green leaves, and the structural color of this feathered crest is advantageous for bird signaling as well. Forested environments (flowers, leaves, light, and shadow) can be advantageous for birds and their signaling as more colorful feathers are found today in tropical regions as demonstrated among extant vibrant coloration of passeriform birds in particular (*Cooney et al., 2022*).

## Materials and methods
### Fossil material
The referred specimen of the enantiornithine bird *Shangyang* sp. (IVPP V 26899) was recovered near La-Ma-Dong Village in Jian-Chang County, within the Jiufotang Formation (dated ~120 Ma). It can be referred to *Shangyang* by the presence of derived features in the premaxillae, tibiotarsus, and sternum (*Figure 1*, Appendix 1).

The feather fragment (labeled sample *Figure 1*) extracted from the specimen was mainly analyzed using SEM (Scanning Electron Microscopy) and S/TEM (Scanning/Transmission Electron Microscopy). The extracted feather sample is one of the longest crest feathers, was displaced far from the caudal position of the skull (sample in *Figure 1*), and belongs to a fully mature asymmetric barb dominant

feather. In addition to the singular feather, three fragments of other feathers were extracted from the in-situ position of the head crest and sampled to validate the revealed structure (Supplementary file 1, *Figure 2—figure supplements 1–3*). A small fragment of the right femur was extracted to produce a ground section to evaluate osteo-histological features in relation to its ontogenetic stage (*Figure 1—figure supplements 1 and 2*).

## Scanning electron microscopy (SEM)

Feather samples coated with gold were examined with a FEI ESEM Quanta 450, which is operated at the Tectonic Laboratory in the China Geological Survey in Beijing, China. The SEM was operated with about 10 mm working distance of the sample under a high voltage of 20 kV. Not only was the surface of the feather sample examined, but the raw material of the cross-section exposed in the ultrathin sliced sections also was imaged to characterize an overall distribution and thickness of soft tissue in comparison with the adjacent sediment or matrix (*Figure 3—figure supplement 1*). Additionally, a FIB-SEM (Heliox 5 CX Dual Beam system) also was further applied to validate the melanosome packing in an extracted volume, and the imaged block is about 11x13 x 9μm (*Figure 3—figure supplement 2*).

## Scanning/transmission electron microscopy

The crest feather fragments were embedded in epoxy resin using the SPI-PON 812 Embedding Kit (MNA, EPOK, DDSA, DMP-30) (*Luft, 1961*) at the Key Laboratory of Vertebrate Evolution and Human Origins of the Chinese Academy of Sciences in Beijing in 2022. After a stepwise polymerization at 37 °C for 12 hr, 45 °C for 12 hr, and 60 °C for 48 hr, the epoxy-embedded block was prepared and cut using an ultrathin slicing machine (Leica EM UC7), equipped with a diamond knife (ultra 45°, 3 mm) for further TEM observations. The ultrathin slices were placed on copper grids with carbon film (200 meshes in ~3 mm diameter copper grid). The thickness of a prepared ultrathin slice is approximately ~70 nm, and the slices were consecutively placed on ~3–5 copper grids. The removed thickness of the sample for making those ~3–5 copper grids for one run of sectioning (serial numbered grids were made) took about ~20 μm of the prepared pyramidal-cubic shaped block (*Figure 3*). The direction of slicing was either perpendicular to the long axis of the barbs or nearly parallel with the long axis of the barb for the two spots selected for cutting.

S/TEM imaging was conducted on a Thermo Scientific (FEI) Talos F200X G2 TEM at the IVPP, Beijing during 2021–2023. The Talos F200X was operated at 200 kV with an analytical scanning transmission electron microscope mode (S/TEM). TEM slices were made from the isolated feather adjacent to the head and rostral portion of the skull. The two specific micron-sized spots of the feather were targeted for examination of their microscopic features by ultrathin slicing and examination as shown in *Figure 3*. For the long contour feather, one spot close to the middle of the feather was made as a cross section of the barb, and for the other spot, the slice was made close to the left side of the fragment with a longitudinal slice (*Figure 3*). The strategic consideration of the two directions (and spots) of cutting was to examine the melanosomes and other features in different views in order to visualize these different features and obtain three-dimensional data (see Appendix 1).

## X-ray computed tomography

The fossil slab was scanned using a GE |phenix| micro-scanner with a resolution of 24.3 μm at Key Laboratory of Vertebrate Evolution and Human Origins, IVPP, Chinese Academy of Sciences, Beijing. For the skeletal anatomy of the bird, μm-CT scans and digital rendering were applied to aid in visualizing the skeletal features of the new specimen (*Figure 1*). The digital rendering of the specimen was completed using Avizo 9.0 (*Figure 1*). Measurements were made using digital calipers, as well as validated in the CT data digitally (*Appendix 1—table 2*). To achieve optimized image resolution, the field of view was adjusted, resulting in part of the feet being excluded from the CT scan. CT scanning data are deposited and available in the open server (https://osf.io/kw7sd/).

## Technical protocol for ultrathin slicing

Prior to the work presented here, we obtained numerous practice sections from a few samples of fossil feathers. In our prior samples, the fragmentary feather tissues were used to improve the technician's capability and precision in sectioning exercises by comparison with the desired direction and sectioned direction for a given fossil. There are several key steps elaborated below for the successful

repetition of practices of sectioning in the correct alignment. First, we examined the fossil feathers with light microscopy and SEM prior to the embedding process to make sure the barbs were aligned correctly in the epoxy resin. Meanwhile, the small part of the extracted feather sample was aligned, using the microscope (Leica UC 7), into a proper position during the embedding and stage trimming process. The attention to feather direction was performed to assure that the barb major extension direction was aligned with the long axis of the bullet-shaped mold (embedding capsule). We also checked the position of the feather fragment in the SPI-Pon 812 during embedding and polymerization. The sliced section should be precise enough to acquire the desired images and information after planning and optimization of each step described above, and before the sectioning. Of course, the ultrathin slicing practices can improve the technician's skill and increase cutting precision, which requires a gradual learning process.

## FDTD modeling of the feather's reflectance

To explore the coloration of the three-dimensionally packed melanosomes, Lumerical finite-difference-time-domain (FDTD) software (Ansys Canada Ltd.) (*Inan and Marshall, 2011*; *Sullivan, 2013*) was used to simulate the visualization of light propagation on the crest feather with regular arrangements of melanosomes (*Figure 1*). The FDTD method allows explicit simulation of light interaction with particular-shaped melanosomes by numerically solving Maxwell's equation in the time domain. The construction of a three-dimensional nanostructure model is based on our characterized data (*Figure 4* and *Appendix 1—table 1*), and detailed modeling steps are included in Appendix 1. The dispersion of the complex refractive index that was applied in our simulation is described and shown in *Figure 4—figure supplement 2*. In order to mimic real visual conditions, several variables were considered during the simulation, including angle of light incidence in two orthogonal incident planes (X-Y and Y-Z plane; *Figure 4b*) and polarization state (s- and p-polarized). Usually, p-polarized light is understood to have an electric field direction parallel to the plane of incidence, and s-polarized light has the electric field oriented perpendicular to the incident plane. Reflectance spectra are presented in *Figure 4c–d*.

## Caveats about potential taphonomic melanosome changes

The simulated colors here are robust even when taking potential taphonomic changes into consideration. The chance that the highly organized and anatomically oriented nanoscale pattern formed here is the result of geological compression or pressure during fossilization and diagenesis is quite low, and such structures are absent or unseen among known taphonomic artifacts (*McNamara et al., 2013*; *Zhao et al., 2020*). At present, we cannot exclude the possibility that shear stress could result in the observed asymmetric packing of melanosomes. However, that pattern has not been reported in any known experiment or geological process of fossilization including those related to Jehol bird or non-avialan dinosaur fossils. Therefore, we suggest the pattern we obtained here represents original biological structures. Given experimental observations of melanosome shrinkage at high pressures and temperatures (~10% shrinkage; *McNamara et al., 2013*; *Zhao et al., 2020*), we also completed additional modeling with the fossil melanosomes enlarged (110%, see *Appendix 1—Tables 3 and 4* and *Figure 4—figure supplement 3*) along with the fossilized keratinous cortex and matrix. This additional modeling resulted in all of the spectral peaks present in the original modeling being recovered. Therefore, our results are robust even when considering known potential taphonomic artifacts, and both approaches produce similar results with the red-to-blue iridescent coloration having been present in the crown feather (*Figure 4—figure supplement 3*).

## Acknowledgements

We thank Yang JM, Ma JB, Wen XL, and Li DS for technical assistance in sampling and ultrathin sectioning and STEM imaging; Hou YM and Yin PF for assistance with micro-CT scanning of the specimen; Gao W for photography; and Xu Y for the artistic reconstruction. Two reviewers are grateful for providing valuable suggestions for improving the earlier version of the manuscript. National Natural Science Foundation of China 42288201(ZZ), National Key Research and Development Project of China (2024YFF0807603) (ZL) and (2024YFF0807602) (YP); NSFC 42472047 (ZL), 37110101 (MY), and NSFC Fund for Excellent Young Scholars KZ 37110101 (MY).

# Additional information

## Funding

| Funder | Grant reference number | Author |
|---|---|---|
| National Science Foundation of China | 42288201 | Zhonghe Zhou |
| National Key Research and Development Program of China | 2024YFF0807603 | Zhiheng Li |
| NSFC Fund for Excellent Young Scholars | KZ 37110101 | Mao Ye |
| National Science Foundation of China | 42472047 | Zhiheng Li |
| National Science Foundation of China | 37110101 | Mao Ye |
| National Key Research and Development Program of China | 2024YFF0807602 | Yanhong Pan |

The funders had no role in study design, data collection and interpretation, or the decision to submit the work for publication.

## Author contributions

Zhiheng Li, Conceptualization, Resources, Data curation, Investigation, Project administration; Jinsheng Hu, Data curation, Software, Formal analysis, Investigation, Visualization, Methodology; Thomas A Stidham, Resources, Investigation, Visualization, Writing – original draft; Mao Ye, Supervision, Funding acquisition, Methodology, Writing – review and editing; Min Wang, Formal analysis, Investigation, Writing – review and editing; Yanhong Pan, Formal analysis, Validation, Methodology, Writing – review and editing; Tao Zhao, Data curation, Visualization, Methodology, Writing – review and editing; Jingshu Li, Resources, Data curation, Writing – review and editing; Zhonghe Zhou, Conceptualization, Supervision, Funding acquisition, Project administration, Writing – review and editing; Julia A Clarke, Data curation, Validation, Methodology, Writing – review and editing

## Author ORCIDs

Zhiheng Li ⓘ https://orcid.org/0000-0002-6968-6982
Jinsheng Hu ⓘ https://orcid.org/0000-0003-0538-4245
Min Wang ⓘ https://orcid.org/0000-0001-8506-1213
Yanhong Pan ⓘ http://orcid.org/0000-0001-5645-1466

Reviewer #1 (Public review): https://doi.org/10.7554/eLife.103628.3.sa1
Reviewer #3 (Public review): https://doi.org/10.7554/eLife.103628.3.sa2
Author response https://doi.org/10.7554/eLife.103628.3.sa3

# Additional files

## Supplementary files

MDAR checklist

## Data availability

All data are provided in the manuscript and the Appendix.

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

## Appendix 1

### Systematic paleontology
Aves *Linné and Salvius, 1758*

Pygostylia *Chiappe and Walker, 2002*
Ornithothoraces *Chiappe, 1995*
Enantiornithes *Walker, 1981*
Genus *Shangyang Wang and Zhou, 2019*

### Brief description and diagnosis of *Shangyang* specimen

The small enantiornithine fossil specimen (IVPP V26899) described here can be referred to *Shangyang* sp. based on the derived skeletal anatomy shared with the holotype (IVPP V25033). For example, the presence of fused premaxillae along their entire length, a nearly square sternal outline, and the elongate fibula help to diagnose the fossil (*Wang and Zhou, 2019*).

The intermetacarpal space between the major and minor metacarpals is quite narrow (*Figure 1—figure supplement 1*). The minor metacarpal is slightly curved and extends distal to the major metacarpal. The main part of the sternum has a square outline, and the lateral trabecula extends caudally, expanding into a large triangular plate (*Figure 1*). The midline projection of the sternum is stout and extends about the same length as the lateral trabecular process (*Figure 1*). The fibula is long, being over two-thirds of the length of the tibiotarsus. The distal pubis is expanded into a boot shape at its distal end (*Figure 1—figure supplement 1*).

Differing slightly from the holotype, the furcula of IVPP V26899 has a longer hypocleidium. The craniolateral process of its sternum also protrudes less than the holotype, and that might relate to its sub-adult ontogenetic stage based on histological examination of its leg bones. In addition, the size of the newly referred specimen is about 12 cm measured from head to tail, which is close to the size of the holotype (*Figure 1*).

### Locality and horizon

The new fossil specimen was recovered from La-Ma-Dong Village in Jian-Chang County in western Liaoning Province, northeastern China. Based on previous geological research, the fossil bird derives from the Lower Cretaceous Jiufotang Formation with the fossil-bearing strata dated as approximately 120 Ma (*Yu et al., 2021*).

### Summary of enantiornithine feathers including the new specimen

Enantiornithes are the most diverse avialan clade among Mesozoic birds (*Chiappe and Walker, 2002*) with particularly abundant and spectacularly preserved fossil specimens from the Early Cretaceous Jehol lagerstätte in northeastern China (*Zhou et al., 2003*; *Zhou and Zhang, 2007*). Those enantiornithine fossils from the Jehol Group preserve a surprising variety of morphological disparity and diversity not only in their skeletons, but also in their striking plumages across their limbs, heads, and tails (*Wang and Zhou, 2017*). Enantiornithine feather assemblages are represented not only by the densely distributed contour and down feathers that covered their whole bodies that functioned largely in insulation, but also by atypical feather morphologies on their heads and other modifications like elongated rectrices (*Foth and Rauhut, 2020*). This morphological diversity across enantiornithine plumages suggests that they played a major role in display, signaling, and sexual selection (*Wang et al., 2021*). Those hypothesized functions are reinforced by the preservation of ancient melanosome pigments in the microstructure of the fossil feathers that would have resulted in a variety of colors, demonstrating the long evolutionary history of feather coloration in the clade.

The well-developed preserved integumental structures of IVPP V26899 are mainly pigmented in a currently black-brown color within the white-gray matrix, distributed surrounding the whole body (*Figure 1*). This plumage not only includes elongate pennaceous flight feathers associated with its forearm, but also a dense covering of contour feathers (including down feathers) distributed along the forehead, cranium, neck, and caudal vertebral region (*Figure 2*). The feathered crest attached to the forehead and cranium of the new specimen is quite darkly pigmented (*Figure 1*). The erect feather bundles show a striking pattern with gradually increasing shaft (or barb) length towards the caudal end of the skull. The highly projected contour feathers approach the occipital region of the skull, forming a triangular crest (*Figure 1*). In addition, the shallow impressions of the paired rectrices are ribbon-like and associated with the pygostyle (*Figure 1—figure supplement 1*). The

long pennaceous flight feathers (or remiges) are well developed and articulated with the forelimbs, including long primaries and secondaries (*Figure 1*).

The long and well-developed feather crest covers most of the dorsal surface of the skull (*Figure 1*). The crown (or crest) feather barbs (or shaft) gradually increase in length and reach their maximum lengths near the caudal part of the skull. The shaft of the longest crest feather is about 2 cm, even longer than the dorsoventral height of the skull (*Figure 1*). The overall shape of the crown feathers forms a triangular crest that is projected dorsally and caudally. The rostral end of the feathered tract extends rostrally down the forehead along the premaxillae (*Figure 1—figure supplement 1*). The crest feathers on the caudal side are longer than the rostral ones. They are highly projected on the parietal region, and a silt-like fissure appears to be present between the feathered bundles attached to the frontal and the parietal (*Figure 1—figure supplement 1*).

The long contour feathers extending from the caudal part of the head are composed of multiple slightly curved barbs, shortly branching off from the calamus at its base (*Figure 1—figure supplement 1*). Densely packed clusters that comprised elongate oval-shaped melanosomes are visible in the surface and the crack region of the feather barbs when examined with the SEM (*Figure 2*, *Wang et al., 2021*).

## Description of skeletal anatomy

The entire specimen (IVPP V26899) was compressed largely dorsoventrally, except for the laterally preserved skull (*Figure 1—figure supplement 1*). The skull is proportionally long, exceeding the length of the humerus. There are three or four conical-shaped teeth present on the premaxillae, and six teeth on the maxillae. At least eight alveolar pits are recognized on the dentary bone, indicating missing teeth. The dorsal surfaces of premaxillae are straight, and a T-shaped lacrimal bone is present. The external narial opening is silt-like, and the antorbital fenestra has a triangular outline. The frontal bones are crushed laterally, with the left and right sides appearing to be fused (*Figure 1—figure supplement 1*).

There are at least eight or nine cervical vertebrae, and they are all exposed in ventral view (*Figure 1—figure supplement 1*). The lateral edge of the coracoid is expanded slightly. The length of the hypocleidium is about the same length as that of the furcular rami. The lateral trabecula of the sternum is caudally expanded into a triangular process. The humerus has a dorsal supracondylar process. The alular metacarpal (cmI) is fused with the major and minor metacarpals proximally, but distally the two longer metacarpals are separate. The proximal part of the left scapula appears to be broken and incompletely fused during healing from a wound (*Figure 1—figure supplement 1*). The length of the first digit is about half of the length of the carpometacarpus. The total length of the forelimb (humerus +carpometacarpus + manual digit) is shorter than that of the hindlimb (*Appendix 1—table 2*).

The distal end of the fused pubis bears a booted process. The pygostyle is distally tapered, forming a spear shape (*Figure 1—figure supplement 1*). The ilium has a dorsal bulge near its middle portion. The fibula is comparatively longer than that of other enantiornithines, reaching over two-thirds of the tibiotarsus length. Proximally, metatarsals II-IV are fused as in other enantiornithines. Metatarsal III is the longest among metatarsals II-IV, with metatarsal IV slightly shorter than metatarsal II. The ungual (claw) size is large in all four pedal digits (*Figure 1—figure supplement 1*), and the ungual lengths in digits I and II are even longer than those of the respective penultimate phalanges.

Ground sections were acquired from the right side of the femur to assess the osteo-histological features of the bone and its ontogenetic stage. As shown in *Figure 1—figure supplement 2*, long, flat-shaped lacunae are present widely and densely packed throughout the major part of the bone section. Very few secondary osteocytes are present, and parallel-fibered bone tissue is underdeveloped. The flattened osteocyte lacunae dominate the cellular shape, with observable vascular canals connecting different lacunae. Overall, the osteo-histology indicates that the bird was still in an active growth stage at the time of death, suggesting it was in its sub-adult growth phase.

## Microscopic methods and ultrastructural morphology of the feather tissue

### Ultrathin sections: preparation and results

We examined two different micron-sized spots for the feather fragment extracted from the new specimen IVPP V26899 (*Figures 1–3*). The surface size of the prepared cubic block in top view for cross section slicing is about 170 ×120 µm (*Figure 3*, *Figure 3—figure supplement 2*).

The advantages of our multiple selections for targeted sampling adopted here are demonstrated by uncovering different aspects of feather and melanosome ultrastructure. In addition, our consecutive ultrathin sectioning of one spot removed a tissue thickness of approximately 20 µm, which was measured from the lower to upper bound of those slices in a given single position. All of those slices were picked up consecutively with 3–5 serially numbered copper grids with carbon film.

High-resolution TEM/STEM (Transmission Electron Microscope/Scanning Transmission Electron Microscopy) imaging of the well-selected spots of sampled feathers was used to investigate the ultrastructure regarding melanosome geometry and keratinous tissues in the thin sections in different views. Our new microscopic data strongly suggests that soft tissue can be preserved well, and it was revealed further in TEM examination. The detailed features revealed here indicate an exceptional preservation of three-dimensional soft tissues in the slab fossil. Our measurements of the thickness of the feather tissue measured from the top view of the section block are about 10–30 µm, *Figure 3—figure supplement 1* (b-d).

### Crest feather samples

The long crest feather was sectioned in two directions with one series made on the side of a barb longitudinally and another series made across the cross section of the central barb (*Figure 3*). The three-dimensional (3-D) packing of the melanosomes was discerned in both the cross- and longitudinal views, with full details (*Figure 3*). The unique asymmetric-hexagonal-packing (AHP) discovered here defines a dense stack of melanosome assemblage array, with additional hooked linkages among the neighboring melanosomes (*Figure 3*). The novel shapes and packing of the melanosomes played a major role in the reduction of air or keratinous space within the barb, and this may correspond to the unique type of structural coloration modeled below. The tiny hooks identified here in some areas, as extra connections, linked the 'head' and 'tail' portions of the sausage-like melanosomes (*Figure 3*). They may have further increased the abrasion resistance of the barbs. The modified sausage-like melanosomes with novel packing have not been reported from any feathers of extant or extinct birds (*Durrer, 1986*; *Zhang et al., 2010*). Differing from the normal compact hexagonal packing (*Durrer, 1986*), the arrangement of these melanosomes appears to be rotated toward the right-lateral side of the feather barb, major axis of the melanosomes lean leftward (*Figure 1—figure supplement 1*). This novel packing could be acquired in the enantiornithine feather through a modification of the simple rodlet-shaped melanosomes previously reported (*Zhang et al., 2010*).

In addition to the middle portion, we sectioned two additional positions from more proximal sides of the feather barbs (*Figure 2—figure supplement 2*) to validate the melanosome packing. As expected, very dense melanosomes were observed. The hooked structures are less clearly defined because of their smaller size or developmental immaturity. However, the asymmetric packing is similarly present, with the melanosome on the right side packed over the melanosome on the left (i.e., longer axis leans toward left in cross section view). As we argued in the main text, this structure in melanosome hooks may help increase the dimension of the barb size, allowing it to grow thicker and enhancing the structural integrity of the barb. The same pattern also is observed in the cranial barbs sampled from the anterior crest feather, which has a much shorter length at the front of the head (see *Figure 2—figure supplement 3*).

### FIB-SEM 3D tomography

The milled block was extracted from the central portion of the long crest feather sample using a Helios 5 CX Dual-Beam system at the IVPP. Before extraction, the surface of the targeted area of fossil sample was deposited with carbon and tungsten (GIS port) and then milled using gallium ions. The nanomanipulator (Easylift) attached to the sampled block for lift-out and attached to a TEM copper grid for further imaging. The sample was milled 20 nm for each time along the 'z' axis and imaged serially (*Figure 3—figure supplement 2*). Resulting serial images revealed intricate structures, such as linkages and hooks, within the stacks of melanosomes.

## Identification of barbs

The overall morphology of the feather represents a common type of primitive feather with several barbs diverging just above the calamus. The major barbs in the middle appear to be slightly larger and thicker in diameter than the neighboring barbs. No evidence of a rachis or barbules was found in the cross section, meaning that the branched barbs are the major components of the feather tissue preserved. The largest diameter of a barb appears to be over 10 μm, exceeding the diameter of most barbules in living feathers (*Figure 3—figure supplement 1*). Given the overall evidence of the morphology and microscopic data (*Figure 2—figure supplements 1*), we interpret the feather as being an asymmetric downy barb-composed feather, without a stiff central shaft and distal branched barbules. It is composed only of several laterally diverged barbs along the same direction with a fusion near a short calamus at its base (*Figure 3—figure supplement 1*).

## FDTD modeling of the crown feather reflectance

Both cross- and longitudinal-section anatomy of the area of crown feather with particular-shaped melanosomes are shown in *Figure 4—figure supplement 1a-b*, *Vinther et al., 2010*. To simulate the spatial and spectral responses of the crown feather, the three-dimensional FDTD modeling is utilized based on the anatomy results aforementioned (*Figure 4—figure supplement 1a-b*). Additionally, the numerical simulation results related to the angle-, polarization-, and wavelength-dependent reflectance spectrum were obtained. In the simulation, novel hexagonal-packed particular-shaped melanosomes were modeled, and in particular, hexagonally arranged oval-shaped (ellipsoidal) microstructures were employed to simulate the derived arrangement of sausage-like melanosomes as shown in *Figure 4—figure supplement 1c*. The crown feather model consists of multiple layers of particular-shaped melanosomes and keratin wrapped around them (10 layers, each layer contains 10 melanosomes) as shown in *Figure 4—figure supplement 1d*. The lengths of semi-axes of the triaxial ellipsoid are assigned according to the statistics of the SEM images (*Appendix 1—table 1*). The period of hexagonal arrangement (e.g., twice of side length of the hexagon) is selected as 406 nm to achieve similar compactness of the model and SEM images (*Appendix 1—table 1*). Specifically, the ellipsoidal melanosomes are arranged in the simulation region so that some of the adjacent melanosomes were in contact with each other to simulate the hooked linkages as shown in *Figure 3d*. The incident light was set accordingly to the s- and p-polarized of two orthogonal incident planes (X-Y and Y-Z plane), respectively. It penetrates through all of the melanosomes, undergoes near-field interactions (such as scattering, interference, etc.), and the reflected light is collected by the reflectance monitor above the incident light plane.

Melanosome (or melanin) is a strongly absorbing pigment, and hence has a complex refractive index ($\bar{n}_m = n_m + i\kappa_m$), in which the real component corresponds to the familiar refractive index of non-absorbing matter and the imaginary part accounts for light absorption (*Wilts, 2013*). For all simulations, we used published values (*Leertouwer et al., 2011*; *Eliason and Shawkey, 2012*; *Eliason and Shawkey, 2012*; *Wilts, 2013*; *Wilts, 2013*; *Stavenga et al., 2015*; *Jiang et al., 2018*) for the refractive indices of keratin ($n_k$) and special melanosomes ($n_m$). The wavelength dependence of the real part and imaginary part of refractive indices is described by Cauchy equation and exponential function as follows:

$$n\left(\lambda\right) = A + B/\lambda^2 \tag{1}$$

$$k\left(\lambda\right) = a \times exp\left(-\lambda/b\right) \tag{2}$$

For concentrated melanin $A_m = 1.677$ and $B_m = 2.81 \times 10^4$ nm², as for extinction coefficient of melanin, $a_m = 0.56$ and $b_m = 270$ nm. In addition, for the real part of the refractive index of keratin, $A_k = 1.532$ and $B_m = 5890$ nm², in all simulations, the imaginary component of keratin is assumed to be negligible (*Wilts, 2013*). In summary, the dispersion of complex refractive index utilized in our simulation is described in *Figure 4—figure supplement 2*.

Non-uniform mesh grids were utilized to obtain a trade-off between sufficient simulation accuracy and simulation time. In addition, we modeled only parts of the crown feather (one hundred novel-packed melanosomes with particular shapes) to reduce computational resources. Stabilized and perfect matched layers (stabilized PMLs) are applied in all directions to avoid boundary reflection. The simulations were performed on a workstation. The processor of the workstation is two Intel Xeon Platinum 8280; one of each containing 28 cores and 56 threads, and each core has a base frequency of 2.70 GHz. For a single simulation running (e.g. computation for single wavelength, one

polarization state (s- or p-polarized), and one incident angle in particular incident plane (X-Y or Y-Z) combination) takes the memory of approximately 130 GB, and the simulation time of about 12 hr.

## Additional FDTD modeling

Taking into consideration the effect of melanosome shrinkage observed in feather melanosome taphonomic experiment under various temperatures and pressures, we performed additional FDTD simulations to account for the melanosome size change (enlarged melanosomes used as shown in *Appendix 1—table 3*). The simulated reflectance spectra and comparison with the original simulation are shown in *Figure 4—figure supplement 3* and *Appendix 1—table 4*. It should be noted that the wavelength of the reflection peak in the 10% increased melanosome size setting (compared to the original measurements) while considering taphonomic change of the melanosomes is almost identical to the original simulation under different incident light. However, the bandwidth of the reflectance peak decreased dramatically (i.e., the reflectance peaks look sharper), which means the feather with enlarged melanosomes should look more strikingly colorful than the normal ones.

**Appendix 1—table 1.** Measurements of the hexagonal-packed ellipsoidal melanosome modeling obtained from partial cross- and longitudinal STEM images (see *Figure 4—figure supplement 1*).

| Parameters | Mean | Standard deviation | Hexagonal period |
|---|---|---|---|
| Length (x) | 186.17 nm | 33.53 nm | |
| Length (y) | 283.49 nm | 46.44 nm | |
| Length (z) | 1774.03 nm | 269.21 nm | 406 nm |

**Appendix 1—table 2.** Measurements of major bone elements in IVPP V26899 (in mm).

| Skull length | | **30.8** | | |
|---|---|---|---|---|
| Humerus length | left | 22.6 | right | 22.2 |
| Ulna length | left | 23.6 | right | 24.4 |
| Scapula length | left | 19.2 | | |
| Radius length | left | 23.6 | right | 23.3 |
| Metacarpal II length | left | 11.5 | | |
| Femur length | left | 21.2 | | |
| Metatarsal III length | left | 15.0 | right | 15.5 |
| Tibiotarsus length | left | 27.0 | right | 27.5 |
| Sternum midline length | | 13.8 | | |
| Pubis length | | 17.2 | | |
| Ilium length | | 13.1 | | |
| Ischium length | | 6.5 | | |
| Pygostyle length | | 8.5 | | |

**Appendix 1—table 3.** Melanosome modeling employed in additional FDTD simulation (enlarged 10% compared to Appendix 1—table 1).

| Parameters | Mean | Standard deviation | Hexagonal period |
|---|---|---|---|
| Length (x) | 206.65 nm | 37.19 nm | |
| Length (y) | 314.67 nm | 51.55 nm | |
| Length (z) | 1969.14 nm | 298.62 nm | 446 nm |

**Appendix 1—table 4.** Reflectance peak wavelengths at different incident angles for the normal and enlarged cases.

| Incident angle (degree) | Reflectance peak wavelength in original sets (nm) | Reflectance peak wavelength at 110% sized case (nm) |
|---|---|---|
| 30 | 875.76 | 893.94 |
| 40 | 681.82 | 693.94 |
| 50 | 578.79 | 584.85 |
| 60 | 518.18 | 519.20 |
| 70 | 475.76 | 475.60 |

