## [Editor Report · eLife Assessment]

This study presents a potentially **fundamental** analysis of a fossil feather from a 120-million-year-old enantiornithine bird. Using sophisticated 3D microscopic and numerical methods, the authors conclude that the feather was iridescent and brightly colored, possibly indicating that this was a male bird that used its crest in sexual displays. At present, the strength of evidence supporting the conclusions is considered **incomplete** based on methodological shortcomings and questions about taphonomy.

---

## [Referee Report · Reviewer #1 (Public review)]

Summary:

Li et al describe a novel form of melanosome based iridescence in the crest of an Early Cretaceous enantiornithine avialan bird from the Jehol Group.

This is an interesting manuscript that describes never before seen melanosome structures and explores fossilised feathers through new methods. This paper creates an opening for new work to explore coloration in extinct birds.

Strengths:

A novel set of methods applied to the study of fossil melanosomes.

Comments on revised version:

The authors provided a response to the previous 9 issues, for which additional response is provided here:

(1) I respectfully disagree with the authors justification regarding the crest. They show one specimen of Confuciusornis with short feathers (which appears to be a unique feature of this species, possibly related to the fact it is beaked) but what about the more primitive Eoconfuciusornis, a referred specimen of which superficially has an enormous "crest" (Zheng et al 2017), as does Changchengornis (Ji et al 1999). Regardless, it would make more sense compare this new specimen to other enantiornithines. Although limited by the preservation of body feathers, which is not all that common, the following published enantiornithines also exhibit a "crest": bohaiornithid indet. (Peteya et al 2017); Brevirostruavis (Li et al 2021); Dapingfangornis (Li et al 2006); Eoenantiornis (Zhou et al 2005); Grabauornis (Dalsatt etal 2014); Junornis (Liu et al 2017); Longirostravis (Hou etal 2004); Monoenantiornis (Hu & O'Connor 2016); Neobohaiornis (Shen etal 2024); Orienantiornis (Liu etal 2019); Parabohaironis (Wang 2023); Parapengornis (Hu etal 2015); Paraprotopteryx (Zheng et al 2007); and every specimen of Protopteryx. In fact, every single published enantiornithine that preserves any feathering on the head has the feathers preserved perpendicular to the bone (in fact, the body feathers on all parts of the bed are splayed at a right angle to the bone due to compression), as shown in the confuciuornis specimen image provided by the authors. Since it is highly improbable they all had crests, the authors have no justification for the interpretation that this new specimen was crested. This does not mean that the feathers were not iridescent or take away from the novel methods these authors have used to explore preserved feathers.

(2) Yes, this is possible, but see above for the very strong argument against interpretation of these feathers as forming a crest.

(3) This just further makes the point that the isolated feather is not likely from the head. Since the neck feathers are missing, it is more likely that it is these feathers that have been disarticulated (and sampled) from the neck region rather than from the very complete looking head feathers; this has significant implications with regards to the birds colour pattern.

(4) Thank you for acknowledging taphonomy.

(5) An interesting hypothesis and one I look forward to seeing explored in the future.

(6) Since the compression is in a single direction, in fact it is not reasonable to assume that distortion would be random. One might predict similar distortion, as with the feathers (spread out from the bone at a 90° angle) and bone (crushed), which are all compressed in a single direction. However, I agree that such a consistent discovery suggests it is not an artifact of preservation, and only further studies will elucidate this

(7) I still fail to detect this hexagonal pattern - could machine learning be used to quantify this pattern? The random arrangement of white arrows does little to clarify the authors interpretations.

(8) Great to see additional sampling

(9) Thank you for the explanation.

---

## [Referee Report · Reviewer #3 (Public review)]

Summary:

The paper presents an in-depth analysis of the original colour of a fossil feather from the crest of a 125-million-year-old enantiornithine bird. From its shape and location, it would be predicted that such a feather might well have shown some striking colour and pattern. The authors apply sophisticated microscopic and numerical methods to determine that the feather was iridescent and brightly coloured, and possibly indicates this was a male bird that used its crest in sexual displays.

Strengths:

The 3D micro-thin-sectioning techniques and the numerical analyses of light transmission are novel and state of the art. The example chosen is a good one, as a crest feather likely to have carried complex and vivid colours as a warning or for use in sexual display. The authors correctly warn that without such 3D study feather colours might be given simply as black from regular 2D analysis, and the alignment evidence for iridescence could be missed.

Weaknesses: Trivial

---

## [Author Response]

The following is the authors’ response to the original reviews

**Public Reviews:**

**Reviewer #1 (Public review):**
Summary:Li et al describe a novel form of melanosome based iridescence in the crest of an Early Cretaceous enantiornithine avialan bird from the Jehol Group.Strengths:Novel set of methods applied to the study of fossil melanosomes.Weaknesses:(1) Firstly, several studies have argued that these structures are in fact not a crest, but rather the result of compression. Otherwise, it would seem that a large number of Jehol birds have crests that extend not only along the head but the neck and hindlimb. It is more parsimonious to interpret this as compression as has been demonstrated using actuopaleontology (Foth 2011).

Firstly, we respectfully acknowledge the reviewer’s interpretation.

However, the new specimen we report here is distinct as preserved from *Confuciusornis* (Foth 2011), which belongs to a different clade and exhibits a differently preserved feather crest of a different shape compared to the species described in this study. Figure 3a Foth 2011, Paläontologische Zeitschrift；the cervical feather is much longer than feather from head region in the specimen the referee talked about; It is quite incompletely preserved and much shorter in proportional length (relative to the skull) than the specimen we sampled (see picture below).

**Author response image 1. sa3fig1:** Our new specimen with well-preserved and the feather crest were interpretated as the originally shaped；the cervical feather is largely absent or very short.

In the new specimen there is a large feather crest that gradually extends from the cranial region of the fossil bird, rather than the cervical region, as observed in the previously proposed *Confuciusornis* crest. The feather crest extends in a consistent direction (caudodistally), and the feathers in the head region of the bird are exceptionally well-preserved, retaining their original shape. The feathers are measured about 1- 2cm at their longest barb. Feathers in the neck are much shorter (see *Confuciusornis* picture above).

(2) The primitive morphology of the feather with their long and possibly not interlocking barbs also questions the ability of such feathers to be erected without geologic compression.

We acknowledge that the specimen must have undergone some degree of compression during diagenesis and fossilization. Given that the rachis itself is already sufficiently thick (that the ligaments everting a crest would attach to), we conclude that it had the structural integrity to remain erect on the skull.

(3) The feather is not in situ and therefore there is no way to demonstrate unequivocally that it is indeed from the head (it could just as easily be a neck feather)

We conclude that it belongs to the head based on the similar suture, overall length, and its close position to the caudal part of the head. There are no similar types of feathers nearby, such as those found on the neck or other areas, which is why we reason that it is a head crest feather. Besides, the shape of the feather we sampled is dramatically different from the much softer and shorter ones detected on the neck.

In addition, we further sampled the crest feather barb from *in situ* preserved feather crest. We also detected a similar pattern to what we originally found regarding the packing of melanosomes. This is now added to the text.

(4) Melanosome density may be taphonomic; in fact, in an important paper that is notably not cited here (Pan et al. 2019) the authors note dense melanosome packing and attribute it to taphonomy. This paper describes densely packed (taphonomic) melanosomes in non-avian avialans, specifically stating, "Notably, we propose that the very dense arrangement of melanosomes in the fossil feathers (Fig. 2 B, C, and G-I, yellow arrows) does not reflect in-life distribution, but is, rather, a taphonomic response to postmortem or postburial compression" and if this paper was taken into account it seems the conclusions would have to change drastically. If in this case the density is not taphonomic, this needs to be justified explicitly (although clearly these Jehol and Yanliao fossils are heavily compressed).

We have added a line acknowledging this possibility. We have accounted for the shrinkage effects caused by heat and compression, as detailed in our Appendix file. Even when these changes are considered, they do not alter the main conclusions of our study. Besides given most melanosomes we used for simulation are mostly complete and well preserved，we consider the distortion is rather limited or at least minor compared to changes seen in taxonomic experiment shown.

(5) Color in modern birds is affected by the outer keratin cortex thickness which is not preserved but the authors note the barbs are much thicker (10um) than extant birds; this surely would have affected color so how can the authors be sure about the color in this feather?

In extant birds, feather barbs of similar size are primarily composed of air spaces and quasi-ordered keratin structures, largely lacking dense melanosomes. The color-producing barb we have described here does not directly correspond to a feather type in modern birds for comparison. Since there is no direct extant analog to inform the keratin thickness and similar melanosome density, we utilize advanced 3-D FDTD modeling approach to the question of coloration reconstruction, rather than relying on statistical DFA approaches. In additional to packed melanosomes, the external thin keratin cortex layer is also considered for the simulation.

Additionally, even in the thinner melanosome-packed layers of barbules in living birds, iridescent coloration often is observed (e.g., Rafael Maia J. R. Soc. Interface 2009). This further supports the plausibility of our modeling approach and its relevance to understanding coloration in both extinct and extant species.

(6) Authors describe very strange shapes that are not present in extant birds: "...different from all other known feather melanosomes from both extant and extinct taxa in having some extra hooks and an oblique ellipse shape in cross and longitudinal sections of individual melanosome" but again, how can it be determined that this is not the result of taphonomic distortion?

We consistently observed similar hook-like structures not only in this feather but also in feathers from different positions of the crest. We do not believe that distortion would produce such a regular and consistent pattern; instead, distortion likely would result in random alterations, as demonstrated by prior taphonomic experiments.

(7) The authors describe the melanosomes as hexagonally packed but this does not appear to be in fact the case, rather appearing quasi-periodic at best, or random. If the authors could provide some figures to justify this hexagonal interpretation?

To further validate the regional hexagonal pattern, we expanded our sampling to additional sites. We observed similar patterns not only in various regions of the same barb but also across different feathers (see added SI Figures below). This extensive sampling supports the validity of the melanosome patterns identified in our original analysis.

(8) One way to address these concerns would be to sample some additional fossil feathers to see if this is unique or rather due to taphonomy

We sampled additional areas from the same feather as well as feathers from other regions of the head crest. The packing patterns are generally similar with slight variations in size (figure S6).

(9) On a side, why are the feet absent in the CT scan image? "

To achieve better image resolution, the field of view was adjusted, resulting in part of the feet being excluded from the CT scan.

**Reviewer #2 (Public review):**
Summary:The authors reconstructed the three-dimensional organization of melanosomes in fossilized feathers belonging to a spectacular specimen of a stem avialan from China. The authors then proceed to infer the original coloration and related ecological implications.Strengths:I believe the study is well executed and well explained. The methods are appropriate to support the main conclusions. I particularly appreciate how the authors went beyond the simple morphological inference and interrogated the structural implications of melanosome organization in three dimensions. I also appreciate how the authors were upfront with the reliability of their methods, results, and limitations of their study. I believe this will be a landmark study for the inference of coloration in extinct species and how to interrogate its significance in the future.

We thank the referee for these positive comments.

Weaknesses:I have a few minor comments.Introduction: I would suggest the authors move the paragraph on coloration in modern birds (lines 75-97) before line 64, as this is part of the reasoning behind the study. I believe this change would improve the flow of the introduction for the general reader.

We thank the referee for the suggestion, and we made changes accordingly to improve the flow of introduction.

Melanosome organization: I was surprised to find little information in the main text regarding this topic. As this is one of the major findings of the study, I would suggest the authors include more information regarding the general geometry/morphology of the single melanosomes and their arrangement in three dimensions.

We thank the referee for this suggestion. We elaborated on the details of the melanosomes in the results as follows:

Hooks are commonly observed on the oval-shaped melanosomes in cross-sectional views, with two dominant types identified on the dorsal and ventral sides (Figure 3c-d, red arrows). These hooks are deflected in opposing directions, linking melanosomes from different arrays (dorsal-ventral) together. The major axis(y) of the oval-shaped melanosomes (mean = 283 nm) is oriented toward the left side in cross-section, while the shorter axis(x) measures approximately 186 nm (Table S2). In oblique or near-longitudinal sections (Figure 3e-f), the hooked structures’ connections to the distal and proximal sides of neighboring melanosomes are clearly visible (blue arrows, Figure 3f). A similar pattern occurs in two additional regions of interest within the same feather (figure S5). Although the smaller proximal hooks in these sections are less distinct, this may reflect developmental variation during melanosome formation along the feather barb. Significantly smaller hooks were also observed in cross-sections of *in-situ* feather barbs from the anterior side of the feather crest (figure S6). The mean long axis (z) of the melanosomes is approximately 1774 nm (Table S2). Based on these observations, we propose that the hooked structures—particularly those on the dorsal, ventral, proximal, and distal sides of the melanosomes—enhance the structural integrity of the barb (figure S7). However, these features may be teratological and unique to this individual, as no similar structures have been reported in other sampled feathers. These hooks may stabilize the stacked melanosome rods and contribute to increased barb dimensions, such as diameter and length. The sections exhibit modified (or asymmetric) hexagonally packed melanosomes with presence of extra hooked linkages (Figure 3c-d and e-f). The long rod-like melanosomes are different from all other known feather melanosomes from both extant and extinct taxa in having some extra hooks and an oblique ellipse shape in cross and longitudinal sections of individual melanosomes (Durrer 1986, Zhang, Kearns et al. 2010). The asymmetric packing of the melanosomes (the major axis leans leftward) played a major role in the reduction of fossilized keratinous matrix within the barbs, which may correspond to a novel structural coloration in this extinct bird. The close packed hexagonal melanosome pattern found in extant avian feathers yield rounded melanosome outlines in contrast to the oval-shaped melanosomes see figure S8, x.

Added Supplemental figure S5. STEM images of cross-sections taken from three different positions (indicated by white dashed lines in a) demonstrate similar melanosome packing styles. Dashed-lines labeled in (a) indicate where the corresponding position of these sections were taken, black arrows indicate the individual barbs that accumulated together in this long crest father. One distinct feature of these sections is the hooked-link structure that aligns the melanosomes into a modified hexagonal, packed arrangement. White arrows (in c, e, g) indicate the hooked structures observed in the selected melanosomes.

Added Supplemental figure S6. STEM images showing melanosome structure from three fragments of the feather crest (indicated by dashed lines and white box in a) reveal the hooked linkages between melanosomes and their surrounding melanosomes structures in (b), (c) and (d). Due to the shorter length of these feather barbs, the hook structures are not as well-defined as those in the longer feather samples shown in the main text.

Keratin: the authors use such a term pretty often in the text, but how is this inference justified in the fossil? Can the authors extend on this? Previous studies suggested the presence of degradation products deriving from keratin, rather than immaculated keratin per se.

We changed to keratinous matrix and material instead. We observed matrix/material in between these melanosomes were filled by organic rich tissue that is proposed to possibly be taphonomically altered keratin.

Ontogenetic assessment: the authors infer a sub-adult stage for the specimen, but no evidence or discussion is reported in the SI. Can the authors describe and discuss their interpretations?

Thanks for the suggestion. We made an osteo-histological section and add our evaluation of the histology of the femoral bone tissue sampled from the specimen to justify assessment of its ontogenetic stage.

See Supplemental figure S2 for Femur Osteo-Histology

SI file Femur Osteo-Histology

Ground sections were acquired from the right side of the femur to assess the osteo-histological features of the bone and its ontogenetic stage. As shown in figure S2, long, flat-shaped lacunae are widely present and densely packed throughout the major part of the bone section. Very few secondary osteocytes are present, and parallel-fibered bone tissue is underdeveloped. The flattened osteocyte lacunae dominate the cellular shape, with observable vascular canals connecting different lacunae. Overall, the osteo-histology indicates that the bird was still in an active growth stage at the time of death, suggesting it was in its sub-adult growth phase.

CT scan data: these data should be made freely available upon publication of the study.

We will release our CT scanning on an open server (https://osf.io/kw7sd/) along with the final version of the manuscript.

**Reviewer #3 (Public review):**
Summary:The paper presents an in-depth analysis of the original colour of a fossil feather from the crest of a 125-million-year-old enantiornithine bird. From its shape and location, it would be predicted that such a feather might well have shown some striking colour and pattern. The authors apply sophisticated microscopic and numerical methods to determine that the feather was iridescent and brightly coloured and possibly indicates this was a male bird that used its crest in sexual displays.Strengths:The 3D micro-thin-sectioning techniques and the numerical analyses of light transmission are novel and state-of-the-art. The example chosen is a good one, as a crest feather is likely to have carried complex and vivid colours as a warning or for use in sexual display. The authors correctly warn that without such 3D study feather colours might be given simply as black from regular 2D analysis, and the alignment evidence for iridescence could be missed.Weaknesses: Trivial.
**Recommendations for the authors:**

**Reviewer #3 (Recommendations for the authors):**
In a few places, the paper can be strengthened:Dimensionality of study method: In the first paragraph, you set things up (lines 60-62) to say that studies hitherto have been of melanosomes and packing in two dimensions... and I then expect you to say soon after, in the next paragraph, 'Here, we investigate a fossil feather in three dimensions...' or some such, but you don't.You come back to Methods at the end of the Introduction (lines 97-101), but again do not say whether you model the feather in three dimensions or not. Yes, you did - I finally learned at line 104 - you did micro serial sectioning. This needs to shift a long forward into the Introduction.

Thanks for the suggestions, we utilize serial sectioning to get a different view of the microbodies that are proposed to be melanosomes and reconstructed the three-dimensional volume of the melanosomes, as well as the intercalated keratin.

We restructured the introduction and make clear that the three-dimensional data obtained in this study also was used for modeling and in a more anterior position in the text.

In the Results, there are not enough references to images. It's not enough to refer generally to 'Figures 3c-f' [line 133] and then go on to rapidly step through some amazing imagery (text lines 133-146) - you need to add an image citation to each observation so readers can know exactly which image is being described each time.

We elaborated our description of imaging to better describe the melanosomes in our results section. We add the description of the stack of melanosomes as IN Above (reply of Reviewer #2).

The 3D data in Figures 3 and 4 is great and based on huge technical wizardry. The sketch model in Figure 4a is excellent, but could you not attempt an actual 3D block diagram showing the hexagonal arrangement of clusters of aligned melanosomes?

We have also tried FIB -SEM in an additional place for validation of our ultrathin sections data. See the SI file.

Added figure S7. Targeted feather barb block prepared in FIB-SEM, with volume rendering reconstruction based on the acquired sequential cross-sectional images; the volume reconstruction is visualized in the x-y plane (c-cross section view) and in x-z plane (d-sagittal section view).

Modified Figure S8d shows the 3D model of aligned melanosomes. To show the arrangement more clearly, the schematic XY cross-section of the melanosomes 3D model is shown below (also shown in Supplementary Figure S8d).

35: delete 'yield'

Changed

73: 'feather fell' ? = 'feather that has fallen'

Changed

305: excises ? = exercises

Changed